# Construction of Chemically Bonded Interface of Organic/Inorganic g-C_3_N_4_/LDH Heterojunction for Z-Schematic Photocatalytic H_2_ Generation

**DOI:** 10.3390/nano11102762

**Published:** 2021-10-18

**Authors:** Yuzhou Xia, Ruowen Liang, Min-Quan Yang, Shuying Zhu, Guiyang Yan

**Affiliations:** 1Fujian Province University Key Laboratory of Green Energy and Environment Catalysis, Ningde Normal University, Ningde 352100, China; yzxia@ndnu.edu.cn (Y.X.); t1629@ndnu.edu.cn (R.L.); 2Fujian Key Laboratory of Pollution Control & Resource Reuse, College of Environmental Science and Engineering, Fujian Normal University, Fuzhou 350007, China; 3College of Chemistry, Fuzhou University, Fuzhou 350116, China; 4Provincial Key Laboratory of Featured Materials in Biochemical Industry, Ningde Normal University, Ningde 352100, China

**Keywords:** chemically bonded interface, heterojunction, Z-scheme, g-C_3_N_4_/LDH, photocatalytic

## Abstract

The design and synthesis of a Z-schematic photocatalytic heterostructure with an intimate interface is of great significance for the migration and separation of photogenerated charge carriers, but still remains a challenge. Here, we developed an efficient Z-scheme organic/inorganic g-C_3_N_4_/LDH heterojunction by in situ growing of inorganic CoAl-LDH firmly on organic g-C_3_N_4_ nanosheet (NS). Benefiting from the two-dimensional (2D) morphology and the surface exposed pyridine-like nitrogen atoms, the g-C_3_N_4_ NS offers efficient trap sits to capture transition metal ions. As such, CoAl-LDH NS can be tightly attached onto the g-C_3_N_4_ NS, forming a strong interaction between CoAl-LDH and g-C_3_N_4_ via nitrogen–metal bonds. Moreover, the 2D/2D interface provides a high-speed channel for the interfacial charge transfer. As a result, the prepared heterojunction composite exhibits a greatly improved photocatalytic H_2_ evolution activity, as well as considerable stability. Under visible light irradiation of 4 h, the optimal H_2_ evolution rate reaches 1952.9 μmol g^−1^, which is 8.4 times of the bare g-C_3_N_4_ NS. The in situ construction of organic/inorganic heterojunction with a chemical-bonded interface may provide guidance for the designing of high-performance heterostructure photocatalysts.

## 1. Introduction

The worsening environmental pollution and energy crisis caused by the large-scale consumption of fossil fuels has posed a great threat to the sustainable development of mankind. Solar-driven photocatalytic H_2_ evolution is deemed to be a promising approach to meet the challenges, due to the easy accessibility and renewability of solar energy [1,2,3,4,5,6,7]. Since the first report of photoelectrochemical water splitting over a TiO_2_ electrode by Fujishima and Honda, diverse catalysts have been developed for photocatalytic H_2_ evolution, including TiO_2_ [8,9,10,11,12], g-C_3_N_4_ [13,14,15], ZnIn_2_S_4_ [16,17], COFs [18], MOFs [19,20,21] and so on. However, despite much progress being made in the area, the overall efficiency is still much less than satisfactory, owing to some stubborn issues, such as insufficient light harvesting and fast recombination of photogenerated electron–hole pairs.

Over the past decades, the construction of heterostructure, especially for Z-scheme heterostructure, is an effective strategy to mediate these problems [22,23,24,25,26,27,28,29,30,31]. In such a system, the photogenerated electrons and holes are spatially separated to the component with a more negative conduction band (CB) position and the counterpart with more positive valence band (VB) position, respectively. This not only benefits the light utilization and boosts the separation of photogenerated charge carriers, but also maintains the high redox ability of the charge carriers. In order to construct an efficient Z-scheme photocatalyst, it is necessary to design a high-quality junction with matched band structures and Fermi energy levels (E_f_) between two components to drive the migration of electrons–holes, as well as a compactly bonded interface to facilitate the charge transfer across the boundary. Currently, great efforts have been devoted to the design of band-aligned heterojunction systems. Relative mature guiding principles are formed. However, less attention has been paid to the interfacial engineering [32,33,34]. Most of the reported heterojunctions are mixtures of two components with weak interaction through Van der Waals forces. As such, photogenerated electrons and holes are easily accumulated and recombined at the interface.

Very recently, the construction of strong interacted interfaces connected via chemical bonds have provided an insight for optimizing the charge migration in terms of efficiency and accuracy, thereby improving the photocatalytic H_2_ evolution performance. For example, Li et al. reported the synthesis of a Mo-S bonded Z-scheme heterojunction by in situ growth of MoSe_2_ on ZnIn_2_S_4_ nanosheets with an S defect [35]. The MoSe_2_–ZnIn_2_S_4_ demonstrates a greatly improved H_2_ generation rate than pristine ZnIn_2_S_4_. Nevertheless, most of the developed chemical-bond-linked heterostructures are all-inorganic systems, which suffer from limitations such as the difficulty in creating anchoring sites for growing the other semiconductor component, and the instability of the surface coordinative unsaturated atoms [36,37,38]. In this context, inspired by the efficient electron transfer of bio-enzyme systems in nature, which are composed of an inorganic metal center and organic coenzyme through the coordination of metal and protein, the construction of organic/inorganic hybrids may be a more convenient and universal approach to obtain strong interacted heterojunctions. 

g-C_3_N_4_ is expected to be a promising organic support due to a good coordination ability, as well as appropriate band-edge positions and high visible light absorption. Especially, an ultrathin 2D g-C_3_N_4_ NS is rich in pyridine-like nitrogen atoms, which can coordinate with metal precursors via strong nitrogen–metal interaction, yielding a chemical-bonded interface [39,40,41]. As for the inorganic counterpart, we focus on the layered double hydroxides (CoAl-LDHs) due to the large exposure of transition metal atoms. As such, herein, we purposely design and synthesize an organic/inorganic g-C_3_N_4_/CoAl-LDH heterojunction with a chemical-bond-connected interface for efficient photocatalytic H_2_ generation. The structure, morphology and photoabsorption properties of the prepared samples are characterized in detail. Photocatalytic activity test results reveal that the heterojunction composites show a much higher H_2_ evolution rate than pure g-C_3_N_4_ NS under visible light irradiation. Collective photoelectrochemical characterizations disclose that the excellent photocatalytic behavior can be attributed to the intimate interface with a sufficient contact area, which endows the heterojunction with more high-speed channels for the migration of charge carriers.

## 2. Materials and Methods

### 2.1. Materilas

All reagents were analytical grade and used without further purification. Urea, Co(NO_3_)_2_, Al(NO_3_)_3_, and NH_4_F were purchased from Sinopharm Chemical Reagent Co., Ltd. (Beijing, China).

### 2.2. Preparation of Catalyst

Fabrication of g-C_3_N_4_ NS: Bulk g-C_3_N_4_ was firstly synthesized by annealing urea (10 g) at 550 °C for 4 h at the rate of 2.3 °C min^−1^ in air. The obtained sample was ground into power. Then, 1 g bulk g-C_3_N_4_ power was added into 50 mL H_2_O and ethyl alcohol with a volume ratio of 1:1 and sonicated for 2 h. The resultant suspension was centrifuged at 3500 rpm for 10 min to remove the residual layered precursor. Consequently, a g-C_3_N_4_ NS suspension with a concentration of ~2 mg mL^−1^ was obtained, denoted as CN. 

Organic/inorganic g-C_3_N_4_/CoAl-LDH heterojunction was synthesized by in situ growth of CoAl-LDH onto g-C_3_N_4_ NS. Typically, 0.25 mmol Co(NO_3_)_2_, 0.125 mmol Al(NO_3_)_3_, 4 mmol urea and 8 mmol NH_4_F were added into the g-C_3_N_4_ NS suspension (50 mL) under vigorous stirring for 30 min. The mixture was heated at 120 °C in a Teflon-lined autoclave for 12 h, then cooled to room temperature naturally. The obtained precipitation was centrifuged and washed with deionized water several times and dried in a vacuum oven overnight at 60 °C, denoted as CN-CoAl_0.25_. A series of g-C_3_N_4_/CoAl-LDH heterojunctions were prepared by varying the amounts of CoAl-LDH precursors via the same synthesis method, and were named as CN-CoAl_x_. For comparison, pure CoAl-LDH was synthesized via the same method without the addition of g-C_3_N_4_ NS.

### 2.3. Characterization 

The as-prepared samples were characterized by powder X-ray diffraction (XRD) on a Bruker D8 Advance X-ray diffractometer (Bruker AXS GmbH, Karlsruhe, Germany), operated at 40 kV and 40 mA with Ni-filtered Cu K irradiation (λ = 1.5406 Å). The Fourier-transform infrared (FTIR) spectra were carried out on a Nicolet 670 FTIR spectrometric analyzer (Thermo Electron, Waltham, MA, USA). UV-vis diffuse reflectance spectra (UV-vis DRS) were obtained by using a UV-vis spectrophotometer (Varian Cary 500, Varian, CA, America). The morphologies of the products were observed by scanning electron microscopy (FEI Nova NANO-SEM 230 spectrophotometer, Hillsboro, OR, USA). Transmission electron microscopy (TEM) images were obtained using a JEOL model JEM 2010 EX instrument (JEOL, Tokyo, Japan) at an accelerating voltage of 200 kV. X-ray photoelectron spectroscopy (XPS) measurements were carried out by using a VG Scientific ESCA Lab Mark II spectrometer (VG Scientific Ltd., Manchester, UK), equipped with two ultra-high vacuum 6 (UHV) chambers. The binding energies of all tested samples were calibrated by C 1 s at 284.6 eV. BET surface area tests were performed on an ASAP2020M apparatus (Micromeritics Instrument Corp., Atlanta, GA, USA). Electron paramagnetic resonance (EPR) spectroscopic measurements were tested via Bruker A300 EPR spectrometer (Bruker AXS GmbH, Karlsruhe, Germany). Raman spectra were recorded on a Renishaw Raman spectrometer (Renishaw InVia, Gloucestershire, UK) with a laser beam of λ = 325 nm. PL was measured by a fluorophotometer (Edinburgh FLS1000, Edinburgh Instruments, Livingston, UK) with an excitation wavelength of 375 nm.

### 2.4. Electrochemistry Measurement

The working electrode was prepared on fluorine-doped tin oxide (FTO) glass, which was cleaned by sonication in acetone and ethanol for 30 min, and 5 mg of the as-prepared samples were dispersed in 0.5 mL N, N-dimethylformamide under sonication for 2 h. Additionally, 10 μL of slurry was dip coated onto the FTO side with exposed areas of 0.25 cm^2^. The uncoated parts of the FTO electrodes were sealed with epoxy resin. The electrochemical measurements were performed in a conventional three-electrode cell, using a Pt plate and an Ag/AgCl electrode as a counter electrode and reference electrode, respectively. The working electrode was immersed in a 0.2 M Na_2_SO_4_ aqueous solution for 40 s before measurement. The photocurrent measurement was conducted with a CHI650E electrochemical workstation (Chenhua Instruments, Shanghai, China). Electrochemical impedance spectroscopy (EIS) was recorded on a ZENNIUM IM6 electrochemical workstation (Zahner, Germany). A 300 W Xe lamp (PLS-SXE300C, Perfectlight Co., Beijing, China) was used as a light source. 

### 2.5. Evaluation of Photocatalytic Activity

The photocatalytic H_2_ evolution activity was evaluated in a Pyrex top-irradiation-type reaction vessel connected to a glass-closed gas circulation system. In the typical photocatalytic experiment, 40 mg photocatalyst with 40 μL of 10 mg mL^−1^ H_2_PtCl_6_·6H_2_O were added into 50 mL solution with 10% triethanolamine (TEOA), which acted as sacrificial agent to trap the photogenerated holes. The suspension was vacuum treated for 30 min to eliminate the air. A 300 W Xenon lamp (PLS-SXE300C, Perfectlight Co., Beijing, China) equipped with a 420 nm cut-off filter was used as the light source. H_2_ was detected using an online gas chromatograph (Tianmei, TCD, Ar carrier, Shanghai, China). 

## 3. Results and Discussion

Figure 1a schematically illustrates the synthesis of the organic/inorganic g-C_3_N_4_/CoAl-LDH heterojunction. Bulk g-C_3_N_4_ was ultrasonically exfoliated into 2D nanosheets, firstly. Then, the nanosheets were employed as an organic support to mix with the LDH precursor for hydrothermal treatment, which enabled the in situ growth of CoAl-LDH onto the CN, forming 2D/2D heterojunction. Figure 1b depicts the XRD patterns of the as-prepared samples. Two typical diffraction peaks at 13° and 27° were observed in CN, associated with the trigonal N linkage of tri-s-triazine motifs (100) and periodic stacking of layers for conjugated aromatic systems (001), respectively. For pure CoAl-LDH, all of the peaks can be well indexed to a hexagonal CoAl-LDH phase (JCPDS NO. 51-0045). The diffraction peaks at 11.5°, 23.3°, 34.4°, 38.9° and 46.5° corresponded to the (003), (006), (012), (015) and (018) lattice planes of CoAl-LDH, respectively. In case of CN-CoAl_x_ samples, both characteristic peaks of CN and CoAl-LDH were observed. The peak intensity of LDH increased gradually with the increment in CoAl-LDH content, demonstrating the successful integration of the two components in the heterojunction. 

The chemical structure of the nanocomposites was investigated by FTIR analysis, as shown in Figure 1c. In comparison with CN, the FTIR spectra of CN-CoAl_x_ showed similar characteristic peaks in the range of 900–1700 cm^−1^ assigned to the stretching vibrations of tri-s-triazine heterocyclic rings [42], validating the preservation of the major chemical structure of g-C_3_N_4_ in the heterojunction. As can be seen from the enlarged inset in Figure 1c, the peak at 807 cm^−1^ ascribing to the breathing vibration of tri-s-triazine showed a gradual shift towards a higher wavenumber, with the increasing amount of CoAl-LDH in the composites. The corresponding peak shifted to 809 cm^−1^ for CN-CoAl_0.7_ suggested a strong chemical interaction between CN and CoAl-LDH in the heterojunction.

Figure 2 shows the SEM and TEM images of the CN, CoAl-LDH and CN-CoAl_0.5_, which were employed to analyze the micromorphology and structure information of the samples. As shown in Figure 2a, pure CN exhibited a typical 2D layered structure composed of ultrathin nanosheets. CoAl-LDH (Figure 2b) displayed a nanoflower assembly structure of nanosheets. As for CN-CoAl_0.5_, the characteristic nanosheet structure of CN was observed (Figure 2c). However, no obvious LDH nanoflower could be detected. This might have been caused by the CN nanosheets that assisted the in situ growth of the CoAl-LDH, which promoted the dispersion of the LDH and inhibited its aggregation. When it further increased the loading amount of CoAl-LDH onto g-C_3_N_4_, both the characteristic structure of CN and CoAl-LDH were observed (Appendix A). The excessive loading of the CoAl-LDH caused the self-aggregation. Figure 2d–f displays the TEM images of the CN-CoAl_0.5_. It is obvious that the heterojunction displayed a sheet-on-sheet structure with planar interface, which is conductive for the high flow and fast transference of charges, due to the large contact interface and excellent electron mobility [43]. HRTEM analysis revealed obvious lattice fringes of 0.26 nm that are assigned to the (012) facets of CoAl-LDH [44]. Meanwhile, an amorphous CN was observed to attach closely to the CoAl-LDH, suggesting a good interfacial contact between the two components. EDS elemental mappings of CN-CoAl_0.5_ presented the co-existence of C, N, Co, Al and O, further validating the formation of a heterojunction structure.

Moreover, the composition and chemical state of the as-prepared samples were measured by XPS. Both elements of CN and CoAl-LDH existed in the CN-CoAl_0.5_ (Figure 3a), verifying the integration of CN with CoAl-LDH. For the C 1 s spectrum of CN (Figure 3b), two peaks located at the binding energies of 284.6 and 288.0 eV were detected, which corresponded to the sp^2^ C-C and N-C=N units, respectively. In comparison, the C 1 s of CN-CoAl_0.5_ could be fitted into three peaks at 284.6, 287.9 and 289.5 eV. The new peak at 289.5 eV could be attributed to C=O, which was generated from the hydrolysis of urea in the synthesis of CoAl-LDH [44]. The N 1 s spectrum of CN-CoAl_0.5_ could be fitted into three main peaks at 398.5, 399.9 and 401.0 eV (Figure 3c), which were assigned to sp^2^-hybridized nitrogen (C-N=C), tertiary nitrogen N-(C)_3_ and free amino units (C-N-H), respectively [45]. The N 1s of CN was similar with that of CN-CoAl_0.5_, excepting for a shift of the peak responding to C-N=C at 398.4 eV. In Figure 3d, three pairs of peaks were detected for Co 2p spectra in both CN-CoAl_0.5_ and CoAl-LDH. The main peaks at 781.1 and 783.9 eV of CoAl-LDH were assigned to Co^3+^ and Co^2+^, respectively [46]. A slight shift towards low binding energy of Co^2+^ (783.8 eV) was observed for CN-CoAl_0.5_. In addition, the Al 2p spectra of both CN-CoAl_0.5_ and CN were located at 74.0 eV (Figure 3e), confirming the Al^3+^ in the samples. The O 1 s spectra of CN-CoAl_0.5_ and CN also showed no difference (Figure 3f). The predominant peak at 531.4 eV was attributed to the lattice oxygen, while the peak at 533.4 eV was assigned to the chemisorbed oxygen [47]. Thus, it is notable that there was an increase in the binding energy of N and decrease in Co^2+^ in the CN-CoAl_0.5_ sample, as compared to that in CN and CoAl-LDH, while other elements showed analogous chemical states. This result suggests that a strong interaction between CoAl-LDH and CN was formed through the coordination of Co with N in the heterojunction.

The chemical structure of the CN-CoAl heterojunction and the interaction between the CoAl-LDH and CN components was further studied by an EPR and Raman spectra. As shown in Appendix A, bulk g-C_3_N_4_ showed no EPR signal in the g range of 1.92–2.08. When bulk g-C_3_N_4_ was exfoliated into 2D nanosheets, an obvious signal at g = 2.003 for CN was observed (Figure 4a), corresponding to the unpaired electrons in π-bonded aromatic rings caused by C defects [48]. Notably, with the integration with CoAl-LDH, the peak was gradually intensified with the increasing amount of CoAl-LDH, which may be ascribed to the redistribution of π-electrons caused by the strong coordination of N with metal Co. The defect could serve as effective electron “traps” to accelerate the separation of photocarriers [49], thus benefiting the photocatalytic performance. Raman spectra of the as-prepared samples were recorded and are displayed in Figure 4b. The characteristic peaks of pure CN at 481, 592, 766, 874, 978 and 1119 cm^−1^ were assigned to the C-N extended network, consistent with those obtained from pristine CN in the literature [50]. The Raman peak at 707 cm^−1^ arose from the breathing mode of the s-triazine ring in g-C_3_N_4_, while the peak at 664 cm^−1^ was associated with the heptazine ring structure of CN. The CN-CoAl heterojunctions displayed similar spectra as those of pure CN, suggesting the chemical structure of CN was preserved. However, the peak ascribed to the heptazine ring structure at 664 cm^−1^ of CN-CoAl composites exhibited a slightly negative shift when compared to the spectrum of the bare CN, which may have been caused by the formation of a new chemical bond at the interface between g-C_3_N_4_ and CoAl-LDH. The result well matched the FTIR and XPS analyses, consolidating the strong chemical interaction between CN and CoAl-LDH.

Furthermore, the optical absorption properties of the bare CN, CoAl-LDH and CN-CoAl_x_ composites were measured by DRS. As shown in Figure 4c, the absorption edge of CN was around 480 nm, revealing its visible-light response characteristic. CoAl-LDH showed two distinct absorption peaks at 280 nm and in the range of 450–550 nm. The absorption edge at 280 nm was assigned to the ligand-to-metal charge transfer of CoAl-LDH, while the absorption at 530 nm was generated from d-d transitions of Co^2+^ in an octahedral geometry [44,51]. As for the heterojunction composites, both characteristic peaks of the CN and CoAl-LDH were observed, suggesting the good integration of the two components in CN-CoAl_x_. Based on the transformed Kubelka–Munk function plots (Appendix A), the band gaps (E_g_) of the CN and CoAl-LDH were measured to be 2.6 eV and 2.1 eV, respectively. Nitrogen (N_2_) adsorption–desorption measurements were measured to investigate the surface properties of the obtained catalysts. As presented in Figure 4d, all of the measured samples showed a type IV adsorption isotherm, revealing their mesoporous structure. The BET surface area of the CN, CN-CoAl_0.5_ and CoAl-LDH were determined to be 33.4, 25.3 and 18.8 m^2^ g^−1^, respectively. The CN-CoAl_0.5_ showed a moderate surface area, which likely resulted from the hybridization of CN with CoAl-LDH. 

The photocatalytic H_2_ evolution activities of the as-prepared samples are presented in Figure 5a. No H_2_ was detected after 4 h irradiation for pristine CoAl-LDH. In the case of bare CN, a relatively low H_2_ production with the value of 233.2 μmol g^−1^ was detected under visible light irradiation of 4 h, which should have been restricted by the fast recombination of the photogenerated electrons and holes. After coupling with CoAl-LDH, the photocatalytic H_2_ production activities of the composites significantly increased. The optimal CN-CoAl_0.5_ showed the highest H_2_ evolution amount of 1952.9 μmol g^−1^, which is 8.4 times of CN. Moreover, the loading amount of the CoAl-LDH on g-C_3_N_4_ played a significant role in the photocatalytic H_2_ generation performance. With the increase in the CoAl-LDH content from 0.25 mmol to 0.5 mmol in the hybrids, the H_2_ evolution rate enhanced gradually. The boosted photocatalytic activity may be due to the increased interacted interface and surface active sites in the heterojunction, which facilitated the separation of photogenerated carriers and promoted the surface reaction. However, a further increase in CoAl-LDH to 0.7 mmol depressed the H_2_ evolution activity. This may have been caused by the large amount of CoAl-LDH that shielded the light absorption of CN. The excessive loading of CoAl-LDH caused self-aggregation, which decreased the exposure of surface active sites, thus leading to a decline in the photocatalytic performance. XPS measurement of the as-prepared samples was performed to investigate the effect of the Co^2+^/Co^3+^ ratio on photocatalytic activity. As shown in Appendix A, the Co species were in the form of a mixed state of +2 and +3 for all of the hybrid samples. By normalizing the peak areas, the atom ratios of Co^2+^/Co^3+^ were calculated to be 2.11, 2.14, 2.07 and 2.01 for the CN-CoAl_0.25_, CN-CoAl_0.3_, CN-CoAl_0.5_ and CN-CoAl_0.7_, respectively. The Co^2+^/Co^3+^ ratios were almost the same for the different CN-CoAl samples, while their photocatalytic activities varied a lot. The result suggests that the Co^2+^/Co^3+^ ratio was not a main factor in affecting the H_2_-generation activity of the CN-CoAl composites.

For comparison, a reference catalyst of a CN/CoAl-Mix was also prepared by physical mixing of CN with CoAl-LDH. As shown in Appendix A, the H_2_ evolution rate of the CN/CoAl-Mix was slightly higher than the pure CN, while much lower than that of CN-CoAl_0.5_. The result directly proves that the construction of strong chemical bond interacted 2D/2D heterojunction is more effective for boosting the photocatalytic H_2_ production activity. The stability of photocatalytic H_2_ production over the optimized CN-CoAl_0.5_ was also evaluated. As presented in Figure 5b, no obvious decrease in catalytic activity was observed during the five recycle tests. Moreover, the surface area and morphology of the CN-CoAl_0.5_ after five cycles of stability text were investigated and are displayed in Appendix A. The BET surface area was measured to be 28.1 m^2^ g^−1^, which was similar to the value before the reaction. The SEM and TEM images reveal that the 2D/2D sheet-to-sheet structure barely changed. These results verify that the 2D/2D CN-CoAl heterojunction had a satisfactory stability and reusability for photocatalytic H_2_ generation.

To gain mechanistic insight into the enhanced photoactivity of the CN-CoAlx, photoelectric responses of the heterojunction composites were carried out. Figure 6a shows the transient photocurrent test of the samples. The CoAl-LDH-modified CN samples exhibited much higher photocurrent strength compared to the CN, suggesting the constructed 2D/2D heterojunction with a large contact area could efficiently reduce the recombination rate of photogenerated carriers. The electrochemical impedance spectrum (EIS) results reveal that CN-CoAl0.5 exhibited a decreased semicircle compared to the bare CN and CoAl-LDH (Figure 6b), illustrating that the 2D/2D interface strongly favored the migration of photogenerated charge carriers, and thus enhancing the photocatalytic H_2_ evolution activity. The behavior of photogenerated carriers was also monitored via PL spectra. As shown in Figure 6c, pure CN displayed an emission peak at around 470 nm. When the CoAl-LDH was introduced, the heterojunction showed a dramatically depressed PL intensity, suggesting the inhibited recombination of photogenerated electron–hole pairs over the strong interacted heterojunction structure.

Moreover, the specific band structures of the CN and CoAl-LDH were determined by the Mott–Schottky test. As presented in Figure 6d,e, the spectra of both samples under three different constant frequencies showed a positive slope of the line segment, suggesting that the prepared CN and CoAl-LDH were typical n-type semiconductors. The derived values of the flat-band potentials (E_f_) of CN and CoAl-LDH were −1.4 V and −0.73 V (vs Ag/AgCl) at pH 7, respectively. According to the conversion equation of E_NHE_ = E_Ag/AgCl_ + E^0^_Ag/AgCl_, (E^0^_Ag/AgCl_ is about 0.197 V at 25 °C, pH = 7), E_f_ of CN and CoAl-LDH were determined to be −1.20 V and −0.53 V versus the normal hydrogen electrode (NHE, pH = 7). Thereby, the valence band (VB) of CN and CoAl-LDH could be calculated to be 1.40 V and 1.57 V. Furthermore, the Kelvin probe test was measured to study the interfacial electronic structure. Figure 6f depicts the measured contact potential difference (CPDs) of the as-prepared samples related to the Au (5.1 eV) reference. The work function of the CN CoAl-LDH and CN-CoAl_0.5_ were calculated to be 4.62 eV, 5.4 eV and 4.85 eV, respectively. 

Based on the above analysis, a plausible photogenerated charge carrier transfer mechanism over the CN-CoAl heterojunction could be proposed. As shown in Figure 7a, due to the lower Femi level of CoAl-LDH than that of CN, free electrons transferred from CN to CoAl-LDH through the N–metal bond until an equilibrium state formed (Figure 7b). As such, an interfacial electric field oriented in the direction from the CN to CoAl-LDH emerged. Under light irradiation, both CN and CoAl-LDH could be photoexcited to generate electron–hole pairs. With the driving force of the built-in electric field, photogenerated electrons from the CB of CoAl-LDH transferred to the VB of CN and recombined with the holes (Figure 7c), thus realizing the Z-scheme charge transfer. Concurrently, the accumulated electrons on the CB of CN reacted with H_2_O for H_2_ production, while the holes of CoAl-LDH were trapped by TEOA.

## 4. Conclusions

In summary, a 2D/2D organic/inorganic g-C_3_N_4_/CoAl-LDH heterojunction with a strong interacted interface was synthesized via an in situ hydrothermal method. The resulting composites showed a markedly improved photocatalytic H_2_ production activity. The optimal CN-CoAl composite displayed a H_2_ evolution of 1952.9 μmol g^−1^ for 4 h visible light irradiation, which is 8.4 times as that of pure CN. The enhanced photocatalytic activity crucially relied on the well-matched band positions of g-C_3_N_4_ and CoAl-LDH components, as well as the sufficient interfacial contact between them, which greatly benefited the migration and separation of the photogenerated charge carriers.

## Figures and Tables

**Figure 1 nanomaterials-11-02762-f001:**
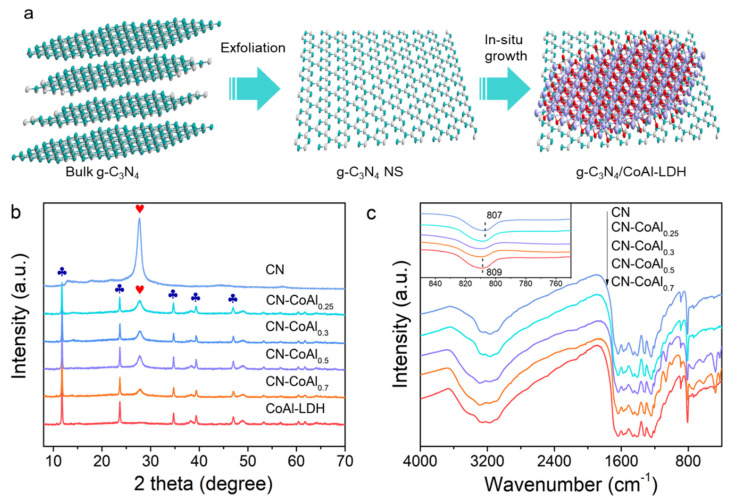
Schematic illustration of preparation of g-C_3_N_4_/CoAl-LDH heterojunction (**a**), XRD patterns (**b**) and FTIR spectra (**c**) of the as-prepared samples.

**Figure 2 nanomaterials-11-02762-f002:**
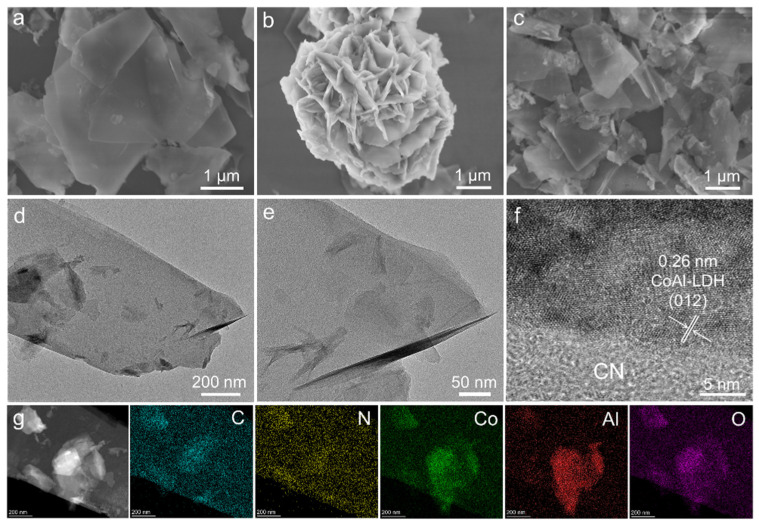
SEM images of CN (**a**), CoAl-LDH (**b**), CN-CoAl_0.5_ (**c**), TEM of CN-CoAl_0.5_ (**d–f**) and the corresponding EDS element mappings (**g**).

**Figure 3 nanomaterials-11-02762-f003:**
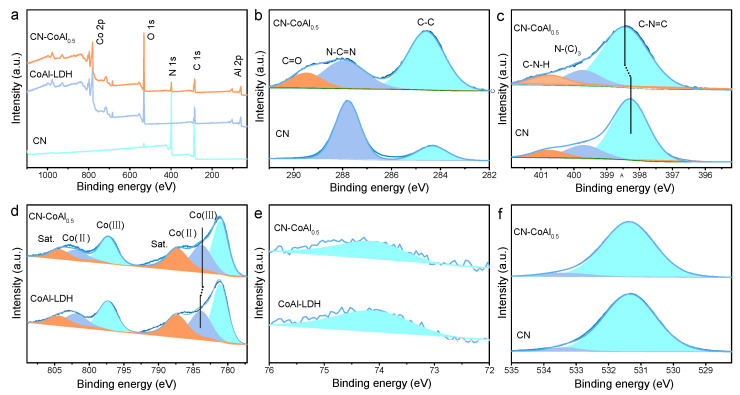
The survey spectra (**a**) and high-resolution XPS spectra of C (**b**), N (**c**), Co (**d**), Al (**e**), O (**f**) of CN-CoAl_0.5_, CN and CoAl-LDH.

**Figure 4 nanomaterials-11-02762-f004:**
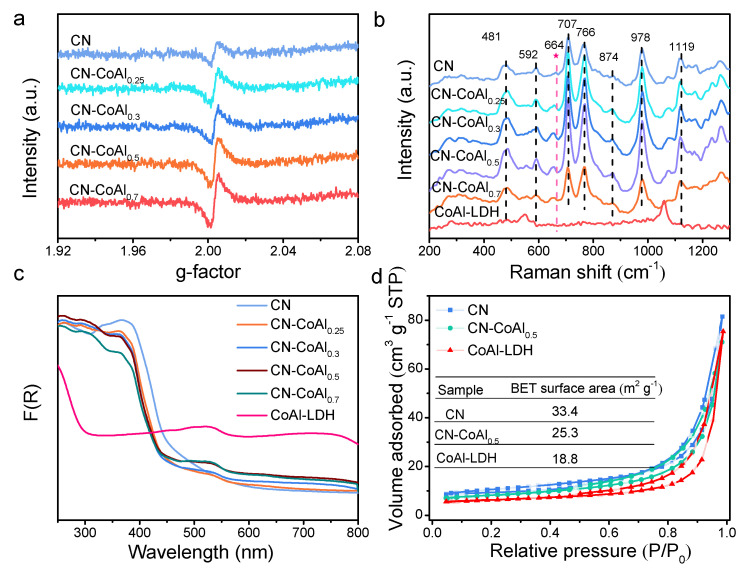
EPR (**a**), Raman (**b**), DRS (**c**) and BET (**d**) analyses of the prepared samples.

**Figure 5 nanomaterials-11-02762-f005:**
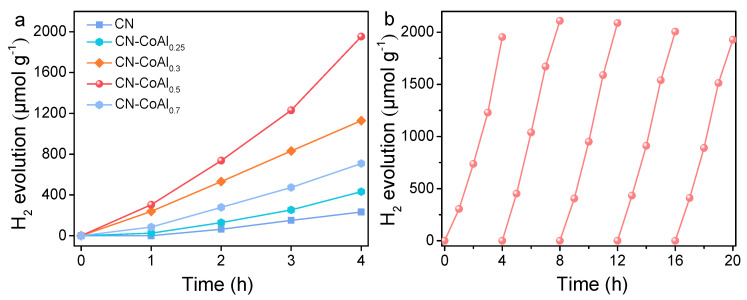
Photocatalytic hydrogen evolution over CN and CN-CoAl_x_ (**a**), stability test of hydrogen evolution over CN-CoAl_0.5_ (**b**).

**Figure 6 nanomaterials-11-02762-f006:**
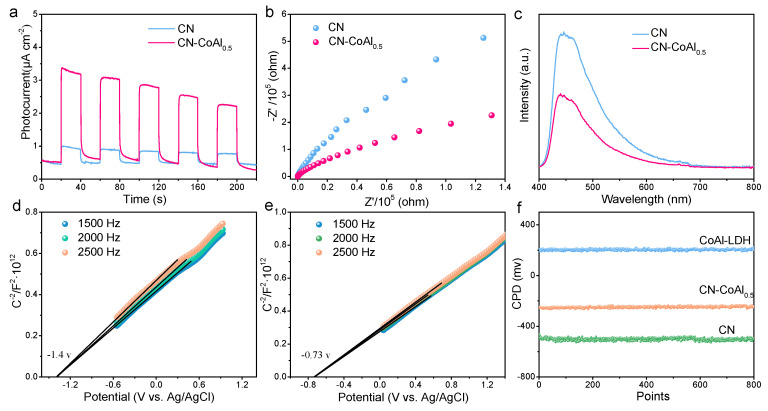
Transient photocurrent response (**a**), EIS (**b**), PL (**c**), Mott–Schottky plots of CN (**d**) and CoAl-LDH (**e**), CPDs of the as-prepared samples surface related to Au reference (**f**).

**Figure 7 nanomaterials-11-02762-f007:**
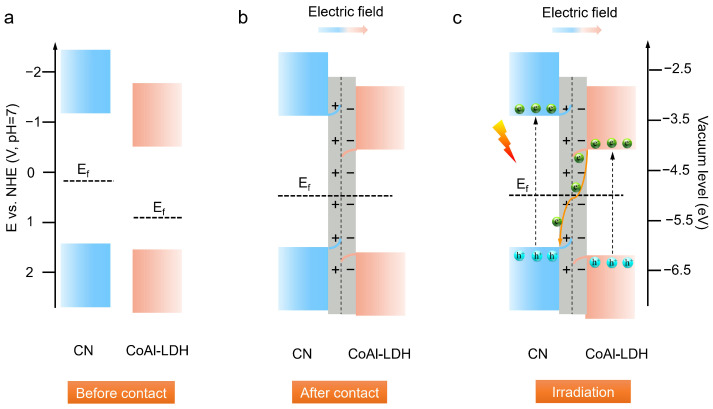
Schematic diagram for band structure of CN and CoAl-LDH before contact (**a**), after contact (**b**) and the light-induced charge transfer from CN to CoAl-LDH toward H_2_ evolution (**c**).

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
