# Peer review of "Construction of Chemically Bonded Interface of Organic/Inorganic g-C3N4/LDH Heterojunction for Z-Schematic Photocatalytic H2 Generation"

_nanomaterials, 2021, doi:10.3390/nano11102762_

Round 1

Reviewer 1 Report

THe paper describes the synthesis of a 2D/2D organic/inorganic g-C3N4/CoAl-LDH heterojunction with 313 strong interacted interface  via a hydrothermal method. It is an interesting option and deserves publication, although the data are not enough discussed under my point of view. 

SOme comments:

Authors have to explain the meaning of "a facile" hydrothermal method.

"the peak at 807 cm-1 in CN ascribed 163 to the breathing vibration of tri-s-triazine units shifts to 809 cm-1 for CN-CoAlx composites, 164 suggesting a chemical interaction between CN and CoAl-LDH" please note that the mentioned shift is not visible in the FTIR figure.

The authors have included a lot of characterization results but they have hardly discussed some of them. For example, some of the Raman bands are not adequately described/assigned. And the XPS deconvolution peaks have not been included. THus, without a real understanding about the characterization data, not a properly discussion can be done. I would recommend to interpret correctly all the data before publication.

Reviewer 2 Report

The authors reported the synthesis of nano-conjugation of g-C3N4 nanosheets with CoAl-LDH nanoparticles that resulted in 2D-2D hetero-junction of the materials. The authors have also performed an adequate materials characterisations. The work presented in the manuscript will be of great interest for readers working in the photocatalytic field and the reported findings may spark further research. Overall, the manuscript is well written can be published after the authors address the following comments. Further check on the English writing is also necessary as some typo errors were spotted in the manuscript.

Comments:

  1. It is interesting that that there is no free standing of CoAl-LDH nanoflowers were observed. The manuscript reported this phenomenon for CN-CoAl05. Was the same observation also observed for higher loading of CoAl-LDH, such as CN-CoAl0.7? 
  2. What causes the photocatalytic activity of CN-CoAl0.7 to decrease significantly? Perhaps the formation of free standing CoAl-LDH nanoflowers could explain this phenomenon? Any SEM and TEM images for this sample would be helpful.
  3. Further detailed explanation should be given to discuss the enhanced catalytic effect of CoAl-LDH and to discuss the trend of LDH loading in Figure 5. 
  4. Would the photocatalytic activity be affected by the thickness of the CoAl-LDH layers? It has been well reported that CoAl-LDH can be easily exfoliated forming thin nanosheets. Would the deposition of these CoAl-LDH nanosheets on CN further enhance the catalytic activity? 
  5. Would the ratio of Co2+/Co3+ affect the catalytic activity? Different LDH loading may result in different Co2+/Co3+ which may affect the electron transfer. 
  6. How about the particle size distribution of the CoAl-LDH?
  7. For comparison purpose, H2 evolution profile for pristine CoAl-LDH should also be presented.
  8. Some characterizations of the material after the stability test should also be presented, e.g. surface area, SEM and TEM. 

Reviewer 3 Report

In this paper, the authors developed an efficient Z-scheme organic/inorganic g-C3N4/LDH heterojunction by in-situ growing of inorganic CoAl-LDH firmly on organic g-C3N4 nanosheet. The so developed g-C3N4 nanosheet offers efficient trap sits to capture transition metal ions due to the two-dimensional (2D) morphology and to the exposition of pyridine-like nitrogen atoms on the surface.

The paper is properly divided in sections and sub-sections but needs some corrections before its publication in the journal.

  • The authors used multiple references in the text, for example 1-6 or 21-30 in the introduction section. They should try to evidence the contribution of the single reference to the discussion;
  • The authors should extend the literature survey. One example of paper to add is 10.3390/catal11050547;
  • The authors should check the language since some errors are present in the text;
  • The data shown in figure 5a evidenced that a correlation between the Al content and the photocatalytic activity is not present. The activity seems to be in this order: CN-CoAl0.5> CoAl0.3> CoAl0.7> CoAl0.25. May the authors better comment this behavior?

The supplementary material cited in the text was not available to the reviewer, so it was impossible to completely review the paper;

Round 2

Reviewer 1 Report

The authors have respond to all my questions and I think that the paper has been improved with the modifications that have been done.

Reviewer 3 Report

In my opinion, the authors improved the manuscript which can be now considered for publication in the journal.